# COT-GAN: Generating Sequential Data
# via Causal Optimal Transport

**Tianlin Xu**
London School of Economics
t.xu12@lse.ac.uk

**Li K. Wenliang**
University College London
kevinli@gatsby.ucl.ac.uk

**Michael Munn**
Google, NY
munn@google.com

**Beatrice Acciaio**
London School of Economics
ETH Zurich
beatrice.acciaio@math.ethz.ch

## Abstract

We introduce COT-GAN, an adversarial algorithm to train implicit generative models optimized for producing sequential data. The loss function of this algorithm is formulated using ideas from Causal Optimal Transport (COT), which combines classic optimal transport methods with an additional temporal causality constraint. Remarkably, we find that this causality condition provides a natural framework to parameterize the cost function that is learned by the discriminator as a robust (worst-case) distance, and an ideal mechanism for learning time dependent data distributions. Following Genevay et al. (2018), we also include an entropic penalization term which allows for the use of the Sinkhorn algorithm when computing the optimal transport cost. Our experiments show effectiveness and stability of COT-GAN when generating both low- and high-dimensional time series data. The success of the algorithm also relies on a new, improved version of the Sinkhorn divergence which demonstrates less bias in learning.

## 1 Introduction

Dynamical data are ubiquitous in the world, including natural scenes such as video and audio data, and temporal recordings such as physiological and financial traces. Being able to synthesize realistic dynamical data is a challenging unsupervised learning problem and has wide scientific and practical applications. In recent years, training implicit generative models (IGMs) has proven to be a promising approach to data synthesis, driven by the work on generative adversarial networks (GANs) [23].

Nonetheless, training IGMs on dynamical data poses an interesting yet difficult challenge. On one hand, learning complex spatial structures of static images has already received significant effort within the research community. On the other hand, temporal dependencies are no less complicated since the dynamical features are strongly correlated with spatial features. Recent works, including [16, 36, 39, 41, 44], often tackle this problem by separating the model or loss into static and dynamic components.

In this paper, we examine training dynamic IGMs for sequential data. We introduce a **new adversarial objective** that builds on optimal transport (OT) theory, and constrains the transport plans to respect *causality*: the probability mass moved to the target sequence at time $t$ can only depend on the source sequence up to time $t$, see [2, 8]. A reformulation of the causality constraint leads to a new adversarial training objective, in the spirit of [21] but tailored to sequential data. In addition, we demonstrate that optimizing the original Sinkhorn divergence over mini-batches causes biased parameter estimation,

and propose the **mixed Sinkhorn divergence** which mitigates this problem. Our new framework, Causal Optimal Transport GAN (COT-GAN), outperforms existing methods on a wide range of datasets from traditional time series to high dimensional videos.

## 2 Background

### 2.1 Adversarial learning for implicit generative models

Goodfellow et al. [23] introduced an adversarial scheme for training an IGM. Given a (real) data distribution $\mu = \frac{1}{N} \sum_{i=1}^{N} \delta_{x^i}$, $x^i \in \mathcal{X}$, and a distribution $\zeta$ on some latent space $\mathcal{Z}$, the generator is a function $g : \mathcal{Z} \to \mathcal{X}$ trained so that the induced distribution $\nu = \zeta \circ g^{-1}$ is as close as possible to $\mu$ as judged by a discriminator. The discriminator is a function $f : \mathcal{X} \to [0, 1]$ trained to output a high value if the input is real (from $\mu$), and a low value otherwise (from $\nu$). In practice, the two functions are implemented as neural networks $g_\theta$ and $f_\varphi$ with parameters $\theta$ and $\varphi$, and the generator distribution is denoted by $\nu_\theta$. The training objective is then formulated as a zero-sum game between the generator and the discriminator. Different probability divergences were later proposed to evaluate the distance between $\mu$ and $\nu_\theta$ [4, 27, 30, 31]. Notably, the Wasserstein-1 distance was used in [5, 6]:

$$\mathcal{W}_1(\mu, \nu) = \inf_{\pi \in \Pi(\mu,\nu)} \mathbb{E}^\pi[\|x - y\|_1], \tag{2.1}$$

where $\Pi(\mu, \nu)$ is the space of transport plans (couplings) between $\mu$ and $\nu$. Its dual form turns out to be a maximization problem over $\varphi$ such that $f_\varphi$ is Lipschitz. Combined with the minimization over $\theta$, a min-max problem can be formulated with a Lipschitz constraint on $f_\varphi$.

### 2.2 Optimal transport and Sinkhorn divergences

The optimization in (2.1) is a special case of the classical (Kantorovich) optimal transport problem. Given probability measures $\mu$ on $\mathcal{X}$, $\nu$ on $\mathcal{Y}$, and a cost function $c : \mathcal{X} \times \mathcal{Y} \to \mathbb{R}$, the optimal transport problem is formulated as

$$\mathcal{W}_c(\mu, \nu) := \inf_{\pi \in \Pi(\mu,\nu)} \mathbb{E}^\pi[c(x, y)]. \tag{2.2}$$

Here, $c(x, y)$ represents the cost of transporting a unit of mass from $x \in \mathcal{X}$ to $y \in \mathcal{Y}$, and $\mathcal{W}_c(\mu, \nu)$ is thus the minimal total cost to transport the mass from $\mu$ to $\nu$. Obviously, the Wasserstein-1 distance (2.1) corresponds to $c(x, y) = \|x - y\|_1$. However, when $\mu$ and $\nu$ are supported on finite sets of size $n$, solving (2.2) has super-cubic (in $n$) complexity [15, 33, 34], which is computationally expensive for large datasets.

Instead, Genevay et al. [21] proposed training IGMs by minimizing a regularized Wasserstein distance that can be computed more efficiently by the Sinkhorn algorithm; see [15]. For transport plans with marginals $\mu$ supported on a finite set $\{x^i\}_i$ and $\nu$ on a finite set $\{y^j\}_j$, any $\pi \in \Pi(\mu, \nu)$ is also discrete with support on the set of all possible pairs $\{(x^i, y^j)\}_{i,j}$. Denoting $\pi_{ij} = \pi(x^i, y^j)$, the Shannon entropy of $\pi$ is given by $H(\pi) := -\sum_{i,j} \pi_{ij} \log(\pi_{ij})$. For $\varepsilon > 0$, the regularized optimal transport problem then reads as

$$\mathcal{P}_{c,\varepsilon}(\mu, \nu) := \inf_{\pi \in \Pi(\mu,\nu)} \{\mathbb{E}^\pi[c(x, y)] - \varepsilon H(\pi)\}. \tag{2.3}$$

Denoting by $\pi_{c,\varepsilon}(\mu, \nu)$ the optimizer in (2.3), one can define a regularized distance by

$$\mathcal{W}_{c,\varepsilon}(\mu, \nu) := \mathbb{E}^{\pi_{c,\varepsilon}(\mu,\nu)}[c(x, y)]. \tag{2.4}$$

Computing this distance is numerically more stable than solving the dual formulation of the OT problem, as the latter requires differentiating dual Kantorovich potentials; see e.g. [13, Proposition 3]. To correct the fact that $\mathcal{W}_{c,\varepsilon}(\alpha, \alpha) \neq 0$, Genevay et al. [21] proposed to use the *Sinkhorn divergence*

$$\widehat{\mathcal{W}}_{c,\varepsilon}(\mu, \nu) := 2\mathcal{W}_{c,\varepsilon}(\mu, \nu) - \mathcal{W}_{c,\varepsilon}(\mu, \mu) - \mathcal{W}_{c,\varepsilon}(\nu, \nu) \tag{2.5}$$

as the objective function, and to learn the cost $c_\varphi(x, y) = \|f_\varphi(x) - f_\varphi(y)\|$ parameterized by $\varphi$, resulting in the following adversarial objective

$$\inf_\theta \sup_\varphi \widehat{\mathcal{W}}_{c_\varphi,\varepsilon}(\mu, \nu_\theta). \tag{2.6}$$

In practice, a sample-version of this cost is used, where $\mu$ and $\nu$ are replaced by distributions of mini-batches randomly extracted from them.

# 3 Training generative models with Causal Optimal Transport

We now focus on data that consists of $d$-dimensional (number of channels), $T$-long sequences, so that $\mu$ and $\nu$ are distributions on the path space $\mathbb{R}^{d \times T}$. In this setting we introduce a special class of transport plans, between $\mathcal{X} = \mathbb{R}^{d \times T}$ and $\mathcal{Y} = \mathbb{R}^{d \times T}$, that will be used to define our objective function; see Definition 3.1. On $\mathcal{X} \times \mathcal{Y}$, we denote by $x = (x_1, ..., x_T)$ and $y = (y_1, ..., y_T)$ the first and second half of the coordinates, and we let $\mathcal{F}^{\mathcal{X}} = (\mathcal{F}_t^{\mathcal{X}})_{t=1}^T$ and $\mathcal{F}^{\mathcal{Y}} = (\mathcal{F}_t^{\mathcal{Y}})_{t=1}^T$ be the canonical filtrations (for all $t$, $\mathcal{F}_t^{\mathcal{X}}$ is the smallest $\sigma$-algebra s.t. $(x_1, ..., x_T) \mapsto (x_1, ..., x_t)$ is measurable; analogously for $\mathcal{F}^{\mathcal{Y}}$).

## 3.1 Causal Optimal Transport

**Definition 3.1.** *A transport plan $\pi \in \Pi(\mu, \nu)$ is called causal if*

$$\pi(dy_t | dx_1, \cdots, dx_T) = \pi(dy_t | dx_1, \cdots, dx_t) \qquad \text{for all } t = 1, \cdots, T-1.$$

*The set of all such plans will be denoted by $\Pi^{\mathcal{K}}(\mu, \nu)$.*

Roughly speaking, the amount of mass transported by $\pi$ to a subset of the target space $\mathcal{Y}$ belonging to $\mathcal{F}_t^{\mathcal{Y}}$ depends on the source space $\mathcal{X}$ only up to time $t$. Thus, a causal plan transports $\mu$ into $\nu$ in a non-anticipative way, which is a natural request in a sequential framework. In the present paper, we will use causality in the sense of Definition 3.1. Note that, in the literature, the term causality is often used to indicate a mapping in which the output at a given time $t$ depends only on inputs up to time $t$.

Restricting the space of transport plans to $\Pi^{\mathcal{K}}$ in the OT problem (2.2) gives the COT problem

$$\mathcal{K}_c(\mu, \nu) := \inf_{\pi \in \Pi^{\mathcal{K}}(\mu, \nu)} \mathbb{E}^{\pi}[c(x, y)]. \tag{3.1}$$

COT has already found wide application in dynamic problems in stochastic calculus and mathematical finance, see e.g. [1, 2, 3, 7, 9]. The causality constraint can be equivalently formulated in several ways, see [8, Proposition 2.3]. We recall here the formulation most well-suited for our purposes. Let $\mathcal{M}(\mathcal{F}^{\mathcal{X}}, \mu)$ be the set of $(\mathcal{X}, \mathcal{F}^{\mathcal{X}}, \mu)$-martingales, and define

$$\mathcal{H}(\mu) := \{(h, M) : h = (h_t)_{t=1}^{T-1}, \ h_t \in \mathcal{C}_b(\mathbb{R}^{d \times t}), \ M = (M_t)_{t=1}^T \in \mathcal{M}(\mathcal{F}^{\mathcal{X}}, \mu), \ M_t \in \mathcal{C}_b(\mathbb{R}^{d \times t})\},$$

where, as usual, $\mathcal{C}_b(\mathbb{X})$ denotes the space of continuous, bounded functions on $\mathbb{X}$. Then, a transport plan $\pi \in \Pi(\mu, \nu)$ is causal if and only if

$$\mathbb{E}^{\pi}\left[\sum_{t=1}^{T-1} h_t(y_{\leq t}) \Delta_{t+1} M(x_{\leq t+1})\right] = 0 \ \text{ for all } (h, M) \in \mathcal{H}(\mu), \tag{3.2}$$

where $x_{\leq t} := (x_1, x_2, \ldots, x_t)$ and similarly for $y_{\leq t}$, and $\Delta_{t+1}M(x_{\leq t+1}) := M_{t+1}(x_{\leq t+1}) - M_t(x_{\leq t})$. Therefore $\mathcal{H}(\mu)$ acts as a class of test functions for causality. Intuitively, causality can be thought of as conditional independence ("given $x_{\leq t}$, $y_t$ is independent of $x_{>t}$"), that can be expressed in terms of conditional expectations. This in turn naturally lends itself to a formulation involving martingales. Where no confusion can arise, with an abuse of notation we will simply write $h_t(y), M_t(x), \Delta_{t+1}M(x)$ rather than $h_t(y_{\leq t}), M_t(x_{\leq t}), \Delta_{t+1}M(x_{\leq t+1})$.

## 3.2 Regularized Causal Optimal Transport

In the same spirit of [21], we include an entropic regularization in the COT problem (3.1) and consider

$$\mathcal{P}_{c,\varepsilon}^{\mathcal{K}}(\mu, \nu) := \inf_{\pi \in \Pi^{\mathcal{K}}(\mu, \nu)} \{\mathbb{E}^{\pi}[c(x, y)] - \varepsilon H(\pi)\}. \tag{3.3}$$

The solution to such problem is then unique due to strict concavity of $H$. We denote by $\pi_{c,\varepsilon}^{\mathcal{K}}(\mu, \nu)$ the optimizer to the above problem, and define the regularized COT distance by

$$\mathcal{K}_{c,\varepsilon}(\mu, \nu) := \mathbb{E}^{\pi_{c,\varepsilon}^{\mathcal{K}}(\mu, \nu)}[c(x, y)].$$

**Remark 3.2.** *In analogy to the non-causal case, it can be shown that, for discrete $\mu$ and $\nu$ (as in practice), the following limits holds:*

$$\mathcal{K}_c(\mu, \nu) \xleftarrow[\varepsilon \to 0]{} \mathcal{K}_{c,\varepsilon}(\mu, \nu) \xrightarrow[\varepsilon \to \infty]{} \mathbb{E}^{\mu \otimes \nu}[c(x, y)],$$

*where $\mu \otimes \nu$ denotes the independent coupling.*

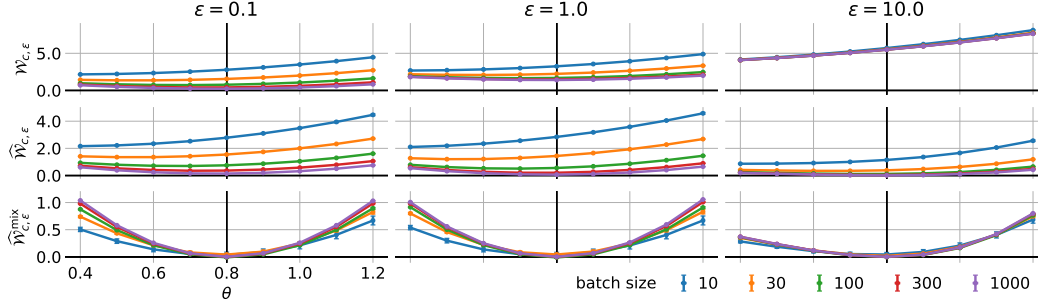

Figure 1: Regularized distance (2.4), Sinkhorn divergence (2.5) and mixed Sinkhorn divergence (3.8) computed for mini-batches of size $m$ from $\mu$ and $\nu_\theta$, where $\mu = \nu_{0.8}$. Color indicates batch size. Curve and errorbar show the mean and sem estimated from 300 draws of mini-batches.

See Appendix A.1 for a proof. This means that the regularized COT distance is between the COT distance and the loss obtained by independent coupling, and is closer to the former for small $\varepsilon$. Optimizing over the space of causal plans $\Pi^{\mathcal{K}}(\mu, \nu)$ is not straightforward. Nonetheless, the following proposition shows that the problem can be reformulated as a maximization over non-causal problems with respect to a specific family of cost functions.

**Proposition 3.3.** *The regularized COT problem* (3.3) *can be reformulated as*

$$\mathcal{P}^{\mathcal{K}}_{c,\varepsilon}(\mu, \nu) = \sup_{l \in \mathcal{L}(\mu)} \mathcal{P}_{c+l,\varepsilon}(\mu, \nu), \tag{3.4}$$

*where*

$$\mathcal{L}(\mu) := \left\{ \sum_{j=1}^{J} \sum_{t=1}^{T-1} h_t^j(y) \Delta_{t+1} M^j(x) : J \in \mathbb{N}, (h^j, M^j) \in \mathcal{H}(\mu) \right\}. \tag{3.5}$$

This means that the optimal value of the regularized COT problem equals the maximum value over the family of regularized OT problems w.r.t. the set of cost functions $\{c + l : l \in \mathcal{L}(\mu)\}$. This result has been proven in [2]. As it is crucial for our analysis, we show it in Appendix A.2.

Proposition 3.3 suggests the following worst-case distance between $\mu$ and $\nu$:

$$\sup_{l \in \mathcal{L}(\mu)} \mathcal{W}_{c+l,\varepsilon}(\mu, \nu), \tag{3.6}$$

as a regularized Sinkhorn distance that respects the causal constraint on the transport plans.

In the context of training a dynamic IGM, the training dataset is a collection of paths $\{x^i\}_{i=1}^{N}$ of equal length $T$, $x^i = (x_1^i, .., x_T^i)$, $x_t^i \in \mathbb{R}^d$. As $N$ is usually very large, we proceed as usual by approximating $\mathcal{W}_{c+l,\varepsilon}(\mu, \nu)$ with its empirical mini-batch counterpart. Precisely, for a given IGM $g_\theta$, we fix a batch size $m$ and sample $\{x^i\}_{i=1}^{m}$ from the dataset and $\{z^i\}_{i=1}^{m}$ from $\zeta$. Denote the generated samples by $y_\theta^i = g_\theta(z^i)$, and the empirical distributions by

$$\hat{\mathbf{x}} = \frac{1}{m} \sum_{i=1}^{m} \delta_{x^i}, \quad \hat{\mathbf{y}}_\theta = \frac{1}{m} \sum_{i=1}^{m} \delta_{y_\theta^i}.$$

The empirical distance $\mathcal{W}_{c+l,\varepsilon}(\hat{\mathbf{x}}, \hat{\mathbf{y}}_\theta)$ can be efficiently approximated by the Sinkhorn algorithm.

### 3.3 Reducing the bias with mixed Sinkhorn divergence

When implementing the Sinkhorn divergence (2.5) at the level of mini-batches, one canonical candidate clearly is

$$2\mathcal{W}_{c_\varphi,\varepsilon}(\hat{\mathbf{x}}, \hat{\mathbf{y}}_\theta) - \mathcal{W}_{c_\varphi,\varepsilon}(\hat{\mathbf{x}}, \hat{\mathbf{x}}) - \mathcal{W}_{c_\varphi,\varepsilon}(\hat{\mathbf{y}}_\theta, \hat{\mathbf{y}}_\theta), \tag{3.7}$$

which is indeed what is used in [21]. While the expression in (3.7) does converge in expectation to (2.5) for $m \to \infty$ ([20, Theorem 3]), it is not clear whether it is an adequate loss given data of fixed batch size $m$. In fact, we find that this is not the case, and demonstrate it here empirically.

**Example 3.4.** *We build an example where the data distribution $\mu$ belongs to a parameterized family of distributions $\{\nu_\theta\}_\theta$, with $\mu = \nu_{0.8}$ (details in Appendix A.3). As shown in Figure 1 (top two rows), neither the expected regularized distance* (2.4) *nor the Sinkhorn divergence* (2.5) *reaches minimum at $\theta = 0.8$, especially for small $m$. This means that optimizing $\nu$ over mini-batches will not lead to $\mu$.*

Instead, we propose the following *mixed Sinkhorn divergence* at the level of mini-batches:

$$\widehat{\mathcal{W}}_{c,\varepsilon}^{\text{mix}}(\hat{\mathbf{x}}, \hat{\mathbf{x}}', \hat{\mathbf{y}}_\theta, \hat{\mathbf{y}}'_\theta) := \mathcal{W}_{c,\varepsilon}(\hat{\mathbf{x}}, \hat{\mathbf{y}}_\theta) + \mathcal{W}_{c,\varepsilon}(\hat{\mathbf{x}}', \hat{\mathbf{y}}'_\theta) - \mathcal{W}_{c,\varepsilon}(\hat{\mathbf{x}}, \hat{\mathbf{x}}') - \mathcal{W}_{c,\varepsilon}(\hat{\mathbf{y}}_\theta, \hat{\mathbf{y}}'_\theta), \qquad (3.8)$$

where $\hat{\mathbf{x}}$ and $\hat{\mathbf{x}}'$ are the empirical distributions of mini-batches from the data distribution, and $\hat{\mathbf{y}}_\theta$ and $\hat{\mathbf{y}}'_\theta$ from the IGM distribution $\zeta \circ g_\theta^{-1}$. The idea is to take into account the bias within both the distribution $\mu$ as well as the distribution $\nu_\theta$ when sampling mini-batches.

Similar to (3.7), when the batch size $m \to \infty$, (3.8) also converges to (2.5) in expectation. So, the natural question arises: for a fixed $m \in \mathbb{N}$, which of the two does a better job in translating the idea of the Sinkhorn divergence at the level of mini-batches? Our experiments suggest that (3.8) is indeed the better choice. As shown in Figure 1 (bottom row), $\widehat{\mathcal{W}}_{c,\varepsilon}^{\text{mix}}$ finds the correct minimizer for all $m$ in Example 3.4. To support this finding, note that the triangular inequality implies

$$\mathbb{E}\left[\left|\mathcal{W}_{c_\varphi,\varepsilon}(\hat{\mathbf{x}}, \hat{\mathbf{y}}_\theta) + \mathcal{W}_{c_\varphi,\varepsilon}(\hat{\mathbf{x}}', \hat{\mathbf{y}}'_\theta) - 2\mathcal{W}_{c,\varepsilon}(\mu, \nu)\right|\right] \leq 2\mathbb{E}\left[\left|\mathcal{W}_{c_\varphi,\varepsilon}(\hat{\mathbf{x}}, \hat{\mathbf{y}}_\theta) - \mathcal{W}_{c,\varepsilon}(\mu, \nu)\right|\right].$$

One can possibly argue that in (3.8) we are using two batches of size $m$, thus simply considering a larger mini-batch in (3.7), say of size $2m$, may perform just as well. However, we found this not to be the case and our experiments confirm that the mixed Sinkhorn divergence (3.8) does outperform (3.7) even when we allow for larger batch size. This reasoning can be extended by considering $\mathcal{W}_{c,\varepsilon}(.,.)$ with more terms for different combinations of mini-batches. In fact, this is what is done in [37], which came to our attention after submitting this paper for review. We have tested different variations in several experiments and while empirically there is no absolute winner, adding more mini-batches increases the computational cost; see Appendix A.3.

### 3.4 COT-GAN: Adversarial learning for sequential data

We now combine the results in Section 3.2 and Section 3.3 to formulate an adversarial training algorithm for IGMs. First, we approximate the set of functions (3.5) by truncating the sums at a fixed $J$, and we parameterize $\mathbf{h}_{\varphi_1} := (h_{\varphi_1}^j)_{j=1}^J$ and $\mathbf{M}_{\varphi_2} := (M_{\varphi_2}^j)_{j=1}^J$ as two separate neural networks, and let $\varphi := (\varphi_1, \varphi_2)$. To capture the adaptedness of those processes, we employ architectures where the output at time $t$ depends on the input only up to time $t$. The mixed Sinkhorn divergence between $\hat{\mathbf{x}}$ and $\hat{\mathbf{y}}_\theta$ is then calculated with respect to a parameterized cost function

$$c_\varphi^{\mathcal{K}}(x, y) := c(x, y) + \sum_{j=1}^J \sum_{t=1}^{T-1} h_{\varphi_1,t}^j(y) \Delta_{t+1} M_{\varphi_2}^j(x). \qquad (3.9)$$

Second, it is not obvious how to directly impose the martingale condition, as constraints involving conditional expectations cannot be easily enforced in practice. Rather, we penalize processes $M$ for which increments at every time step are non-zero on average. For an $(\mathcal{X}, \mathcal{F}^{\mathcal{X}})$-adapted process $M_{\varphi_2}^j$ and a mini-batch $\{x^i\}_{i=1}^m$ ($\sim \hat{\mathbf{x}}$), we define the martingale penalization for $\mathbf{M}_{\varphi_2}$ as

$$p_{\mathbf{M}_{\varphi_2}}(\hat{\mathbf{x}}) := \frac{1}{mT} \sum_{j=1}^J \sum_{t=1}^{T-1} \left| \sum_{i=1}^m \frac{\Delta_{t+1} M_{\varphi_2}^j(x^i)}{\sqrt{\text{Var}[M_{\varphi_2}^j] + \eta}} \right|,$$

where $\text{Var}[M]$ is the empirical variance of $M$ over time and batch, and $\eta > 0$ is a small constant. Third, we use the mixed normalization introduced in (3.8). Each of the four terms is approximated by running the Sinkhorn algorithm on the cost $c_\varphi^{\mathcal{K}}$ for an a priori fixed number of iterations $L$.

Altogether, we arrive at the following adversarial objective function for COT-GAN:

$$\widehat{\mathcal{W}}_{c_\varphi^{\mathcal{K}},\varepsilon}^{\text{mix},L}(\hat{\mathbf{x}}, \hat{\mathbf{x}}', \hat{\mathbf{y}}_\theta, \hat{\mathbf{y}}'_\theta) - \lambda p_{\mathbf{M}_{\varphi_2}}(\hat{\mathbf{x}}), \qquad (3.10)$$

where $\hat{\mathbf{x}}$ and $\hat{\mathbf{x}}'$ are empirical measures corresponding to two samples of the dataset, $\hat{\mathbf{y}}_\theta$ and $\hat{\mathbf{y}}'_\theta$ are the ones corresponding to two samples from $\nu_\theta$, and $\lambda$ is a positive constant. We update $\theta$ to decrease this objective, and $\varphi$ to increase it.

---

**Algorithm 1:** training COT-GAN by SGD

---

**Data:** $\{x^i\}_{i=1}^N$ (real data), $\zeta$ (probability distribution on latent space $\mathcal{Z}$)
**Parameters:** $\theta_0$, $\varphi_0$, $m$ (batch size), $\varepsilon$ (regularization parameter), $L$ (number of Sinkhorn iterations), $\alpha$ (learning rate), $\lambda$ (martingale penalty coefficient)
**Result:** $\theta$, $\varphi$
Initialize: $\theta \leftarrow \theta_0$, $\varphi \leftarrow \varphi_0$
**for** $k = 1, 2, \ldots$ **do**

> Sample $\{x^i\}_{i=1}^m$ and $\{x'^i\}_{i=1}^m$ from real data;
> Sample $\{z^i\}_{i=1}^m$ and $\{z'^i\}_{i=1}^m$ from $\zeta$;
> $(y_\theta^i, y_\theta'^i) \leftarrow (g_\theta(z^i), g_\theta(z'^i))$;
> Compute $\widehat{\mathcal{W}}_{c_\varphi^\mathcal{K},\varepsilon}^{\mathrm{mix},L}(\hat{\mathbf{x}}, \hat{\mathbf{x}}', \hat{\mathbf{y}}_\theta, \hat{\mathbf{y}}_\theta')$ (3.8) by the Sinkhorn algorithm, with $c_\varphi^\mathcal{K}$ given by (3.9);
> $\varphi \leftarrow \varphi + \alpha\nabla_\varphi\Big(\widehat{\mathcal{W}}_{c_\varphi^\mathcal{K},\varepsilon}^{\mathrm{mix},L}(\hat{\mathbf{x}}, \hat{\mathbf{x}}', \hat{\mathbf{y}}_\theta, \hat{\mathbf{y}}_\theta') - \lambda p_{\mathbf{M}_{\varphi_2}}(\hat{\mathbf{x}})\Big)$;
> Sample $\{x^i\}_{i=1}^m$ and $\{x'^i\}_{i=1}^m$ from real data;
> Sample $\{z^i\}_{i=1}^m$ and $\{z'^i\}_{i=1}^m$ from $\zeta$;
> $(y_\theta^i, y_\theta'^i) \leftarrow (g_\theta(z^i), g_\theta(z'^i))$;
> Compute $\widehat{\mathcal{W}}_{c_\varphi^\mathcal{K},\varepsilon}^{\mathrm{mix},L}(\hat{\mathbf{x}}, \hat{\mathbf{x}}', \hat{\mathbf{y}}_\theta, \hat{\mathbf{y}}_\theta')$ (3.8) by the Sinkhorn algorithm, with $c_\varphi^\mathcal{K}$ given by (3.9);
> $\theta \leftarrow \theta - \alpha\nabla_\theta\Big(\widehat{\mathcal{W}}_{c_\varphi^\mathcal{K},\varepsilon}^{\mathrm{mix},L}(\hat{\mathbf{x}}, \hat{\mathbf{x}}', \hat{\mathbf{y}}_\theta, \hat{\mathbf{y}}_\theta')\Big)$;

**end**

---

While the generator $g_\theta : \mathcal{Z} \to \mathcal{X}$ acts as in classical GANs, the adversarial role here is played by $\mathbf{h}_{\varphi_1}$ and $\mathbf{M}_{\varphi_2}$. In this setting, the discriminator, parameterized by $\varphi$, learns a robust (worst-case) distance between the real data distribution $\mu$ and the generated distribution $\nu_\theta$, where the class of cost functions as in (3.9) originates from causality. The algorithm is summarized in Algorithm 1. Its time complexity scales as $\mathcal{O}((J + d)LTm^2)$ for each iteration.

# 4 Related work

Early video generation literature focuses on dynamic texture modeling [17, 38, 42]. Recent efforts in video generation within the GAN community have been devoted to designing GAN architectures of a generator and discriminator to tackle the spatio-temporal dependencies separately, e.g., [36, 39, 41]. VGAN [41] explored a two-stream generator that combines a network for a static background and another one for moving foreground trained on the original GAN objective. TGAN [36] proposed a new structure capable of generating dynamic background as well as a weight clipping trick to regularize the discriminator. In addition to a unified generator, MoCoGAN [39] employed two discriminators to judge both the quality of frames locally and the evolution of motions globally.

The broader literature of sequential data generation attempts to capture the dependencies in time by simply deploying recurrent neural networks in the architecture [19, 24, 29, 44]. Among them, TimeGAN [44] demonstrated improvements in time series generation by adding a teacher-forcing component in the loss function. Alternatively, WaveGAN [16] adopted the causal structure of WaveNet [32]. Despite substantial progress made, existing sequential GANs are generally domain-specific. We therefore aim to offer a framework that considers (transport) causality in the objective function and is suitable for more general sequential settings.

Whilst our analysis is built upon [15] and [21], we remark two major differences between COT-GAN and the algorithm in [21]. First, we consider a different family of costs. While [21] learns the cost function $c(f_\varphi(x), f_\varphi(y))$ by parameterizing $f$ with $\varphi$, the family of costs in COT-GAN is found by adding a causal component to $c(x, y)$ in terms of $\mathbf{h}_{\varphi_1}$ and $\mathbf{M}_{\varphi_2}$. The second difference is the mixed Sinkhorn divergence we propose, which reduces biases in parameter estimation and can be used as a generic divergence for training IGMs not limited to time series settings.

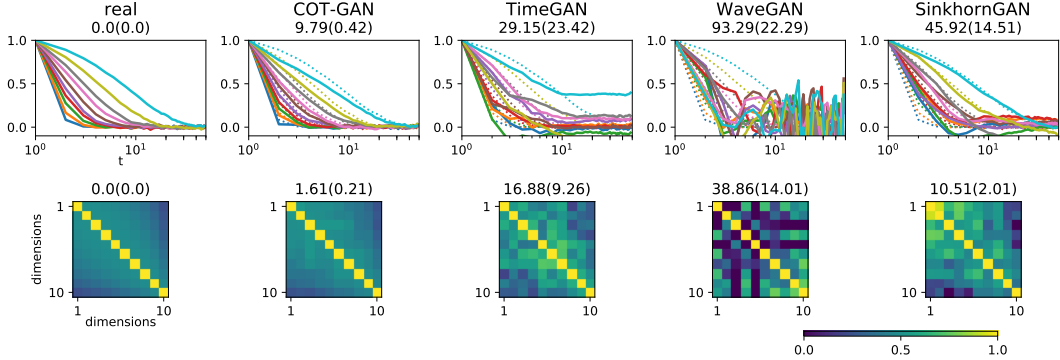

Figure 2: Results on learning the multivariate AR-1 process. Top row shows the auto-correlation coefficient for each channel. Bottom row shows the correlation coefficient between channels averaged over time. The numbers on top of each panel are the mean and standard deviation (in brackets) of the sum of the absolute difference between the correlation coefficients computed from real (leftmost) and generated samples for 16 runs with different random seeds.

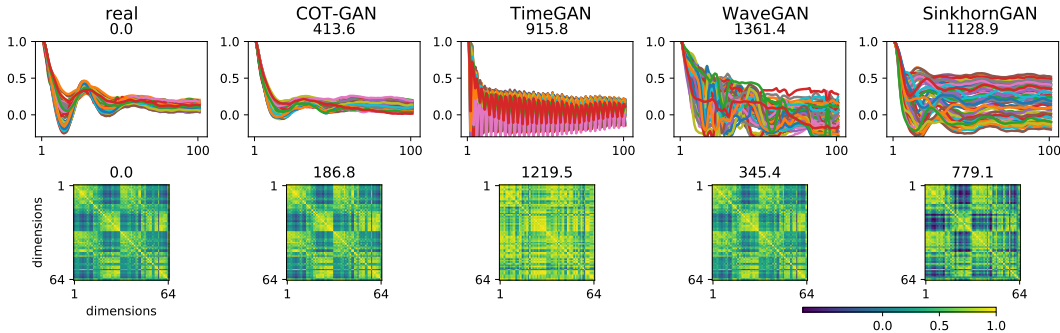

Figure 3: Results on EEG data. The same correlations as Figure 2 are shown.

## 5 Experiments

### 5.1 Time series

We now validate COT-GAN empirically[1]. For times series that have a relatively small dimensionality $d$ but exhibit complex temporal structure, we compare COT-GAN with the following methods: **TimeGAN** [44] as reviewed in Section 4; **WaveGAN** [16] as reviewed in Section 4; and **Sinkhorn-GAN**, similar to [21] with cost $c(f_\varphi(x), f_\varphi(y))$ where $\varphi$ is trained to increase the mixed Sinkhorn divergence with weight clipping. All methods use $c(x, y) = \|x - y\|_2^2$. The networks $h$ and $M$ in COT-GAN and $f$ in SinkhornGAN share the same architecture. Details of models and datasets are in Appendix B.1.

**Autoregressive processes.** We first test whether COT-GAN can learn temporal and spatial correlation in a multivariate first-order auto-regressive process (AR-1).

For these experiments, we report two evaluation statistics: the sum of the absolute difference of the correlation coefficients between channels averaged over time, and the absolute difference between the correlation coefficients of real samples and those of generated samples. We evaluate the performance of each method by taking the mean and standard deviation of these two evaluation statistics over 16 runs with different random seeds.

In Figure 2, we show an example plot of results from a single run, as well as the evaluation statistics aggregated over all 16 runs on top of each panel. COT-GAN samples have correlation structures that best match the real data. Neither TimeGAN, WaveGAN nor SinkhornGAN captures the correlation

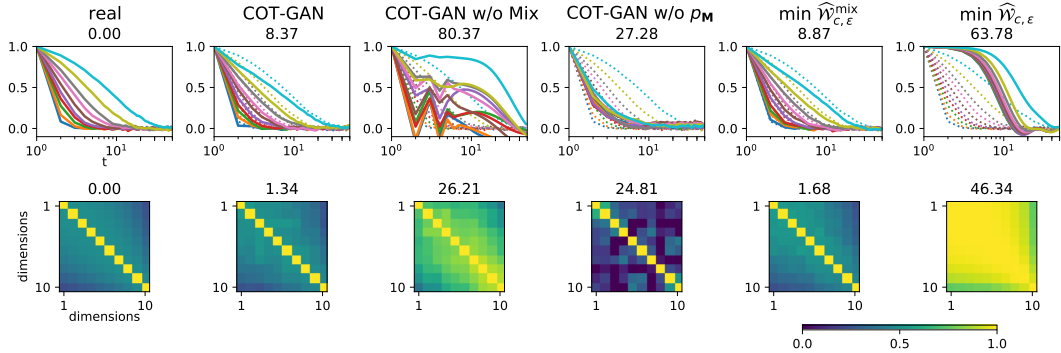

Figure 4: Ablation investigation.

structure for this dataset. The small standard deviation of the evaluation statistics demonstrates that COT-GAN is the most stable model at least in the AR-1 experiment since it produces similar results from each run of the model.

**Noisy oscillations.** The noisy oscillation distribution is composed of sequences of 20-element arrays (1-D images) [43]. Figure 8 in Appendix B.1 shows data as well as generated samples by different training methods. To evaluate performance, we estimate two attributes of the samples by Monte Carlo: the marginal distribution of pixel values, and the joint distribution of the location at adjacent time steps. COT-GAN samples match the real data best.

**Electroencephalography (EEG).** This dataset is from the UCI repository [18] and contains recordings from 43 healthy subjects each undergoing around 80 trials. Each data sequence has 64 channels and we model the first 100 time steps. We compare performance of COT-GAN with respect to other baseline models by investigating how well the generated samples match with the real data in terms of temporal and channel correlations, see Figure 3, and how the coefficient $\lambda$ affects sample quality, see Appendix B.1. COT-GAN generates the best samples compared with other baselines across two metrics.

In addition, we provide an ablation investigation of COT-GAN, in which we study the impact of the components of the model by excluding each of them in the multivariate AR-1 experiment. In Figure 4, we compare the real samples with COT-GAN, COT-GAN using the original Sinkhorn divergence without the mixing, COT-GAN without the martingale penalty $p_{\mathbf{M}}$, direct minimization (without a discriminator) of the mixed and original Sinkhorn divergences from (3.8) and (3.7). We conclude that each component of COT-GAN plays a role in producing the best result in this experiment, and that the mixed Sinkhorn divergence is the most important factor for improvements in performance.

## 5.2 Videos

We train COT-GAN on animated Sprites [28, 35] and human action sequences [12]. We pre-process the Sprites sequences to have a sequence length of $T = 13$, and the human action sequences to have length $T = 16$. Each frame has dimension $64 \times 64 \times 3$. We employ the same architecture for the generator and discriminator to train both datasets. Both the generator and discriminator consist of a generic LSTM with 2-D convolutional layers. Details of the data pre-processing, GAN architectures, hyper-parameter settings, and training techniques are reported in Appendix B.2.

Baseline models chosen for the video datasets are **MoCoGAN** from [39], and direct minimization of the mixed Sinkhorn divergence (3.8), as it achieves a good result when compared to the other methods addressed in Figures 2 and 4. We show the real data and generated samples from COT-GAN side by side in Figure 5. Generated samples from all methods, without cherry-picking, are provided in Appendix C. The evaluation metrics we use to assess model performance are the Fréchet Inception Distance (FID) [25] which compares individual frames, the Fréchet Video Distance (FVD) [40] which compares the video sequences as a whole by mapping samples into features via pretrained 3D convolutional networks, and their kernel counterparts (KID, KVD) [11]. Previous studies suggest

that FVD correlates better with human judgement than KVD for videos [40], whereas KID correlates better than FID on images [46].

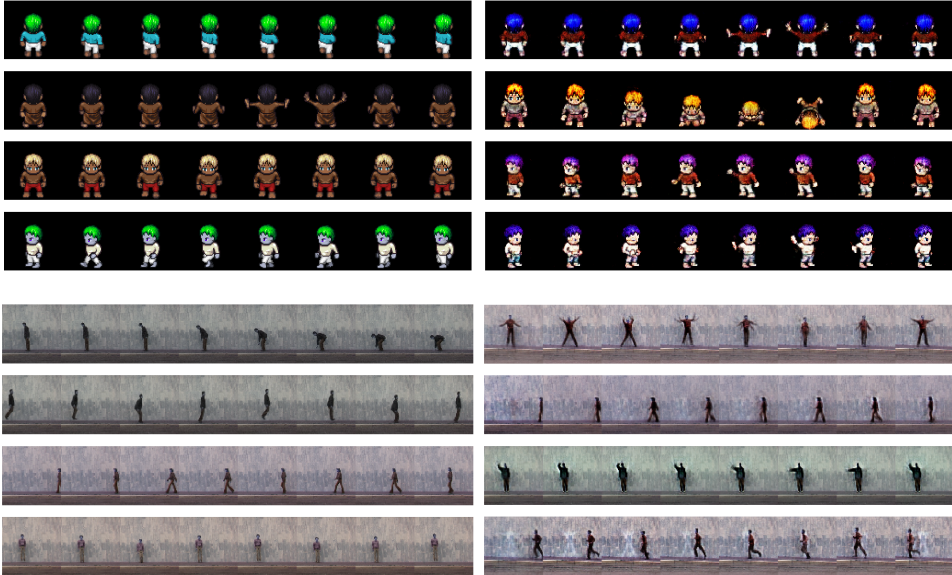

Figure 5: Animated (top) and human (bottom) action videos. Left column reports real data samples, and right column samples from COT-GAN.

Table 1: Evaluations for video datasets. Lower value indicates better sample quality.

| Sprites | FVD | FID | KVD | KID |
|---|---|---|---|---|
| MoCoGAN | 1 108.2 | 280.25 | 146.8 | 0.34 |
| $\min \widehat{\mathcal{W}}_{c,\varepsilon}^{\mathrm{mix}}$ | 498.8 | **81.56** | 83.2 | **0.078** |
| COT-GAN | **458.0** | 84.6 | **66.1** | 0.081 |
| **Human actions** | | | | |
| MoCoGAN | 1 034.3 | 151.3 | 89.0 | 0.26 |
| $\min \widehat{\mathcal{W}}_{c,\varepsilon}^{\mathrm{mix}}$ | 507.6 | 120.7 | **34.3** | 0.23 |
| COT-GAN | **462.8** | **58.9** | 43.7 | **0.13** |

In Table 1 the evaluation scores are estimated using 10,000 generated samples. For Sprites, COT-GAN performs better than the other two methods on FVD and KVD. However, minimization of the mixed Sinkhorn divergence produces slightly better FID and KID scores when compared to COT-GAN. The results in [40] suggest that FID better captures the frame-level quality, while FVD is better suited for the temporal coherence in videos. For the human action dataset, COT-GAN is the best performing method across all metrics except for KVD.

## 6  Discussion

With the present paper, we introduce the use of causal transport theory in the machine learning literature. As already proved in other research fields, we believe it may have a wide range of applications here as well. The performance of COT-GAN already suggests that constraining the transport plans to be causal is a promising direction for generating sequential data. The approximations we introduce, such as the mixed Sinkhorn distance (3.8) and truncated sum in (3.5), are sufficient to produce good experimental results, and provide opportunities for more theoretical analyses in future studies. Directions of future development include ways to learn from data with flexible lengths, extensions to conditional COT-GAN, and improved methods to enforce the martingale property for **M** and better parameterize the causality constraint.

# 7 Broader impact

The COT-GAN algorithm introduced in this paper is suitable to generate sequential data, when the real dataset consists of i.i.d. sequences or of stationary time series. It opens up doors to many applications that can benefit from time series synthesis. For example, researchers often do not have access to abundant training data due to privacy concerns, high cost, and data scarcity. This hinders the capability of building accurate predictive models.

Ongoing research is aimed at developing a modified COT-GAN algorithm to generate financial time series. The high non-stationarity of financial data requires different features and architectures, whilst causality when measuring distances between sequences remains the crucial tool. The application to market generation is of main interest for the financial and insurance industry, for example in model-independent pricing and hedging, portfolio selection, risk management, and stress testing. In broader scientific research, our approach can be used to estimate from data the parameters of simulation-based models that describe physical processes. These models can be, for instance, differential equations describing neural activities, compartmental models in epidemiology, and chemical reactions involving multiple reagents.

## Acknowledgments and Disclosure of Funding

BA thanks the financial support from the Erwin Schrödinger Institute during the thematic programme on Optimal Transport (May 2019, Vienna). This material is based upon work supported by Google Cloud. LKW is supported by the Gatsby Charitable Foundation.

## Footnotes

[1]Code and data are available at github.com/tianlinxu312/cot-gan

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
