[Supplementary Material]

# COT-GAN: Generating Sequential Data via Causal Optimal Transport: Supplementary material

## A  Specifics on regularized Causal Optimal Transport

### A.1  Limits of regularized Causal Optimal Transport

In this section we prove the limits stated in Remark 3.2.

**Lemma A.1.** *Let $\mu$ and $\nu$ be discrete measures, say on path spaces $\mathbb{X}^T$ and $\mathbb{Y}^T$, with $|\mathbb{X}| = m$ and $|\mathbb{Y}| = n$. Then*

$$\mathcal{K}_{c,\varepsilon}(\mu,\nu) \xrightarrow[\varepsilon\to 0]{} \mathcal{K}_c(\mu,\nu).$$

*Proof.* We mimic the proof of Theorem 4.5 in [2], and note that the entropy of any $\pi \in \Pi(\mu,\nu)$ is uniformly bounded:

$$0 \le H(\pi) \le C := m^T n^T e^{-1}. \tag{A.1}$$

This yields

$$
\begin{aligned}
\inf_{\pi\in\Pi^{\mathcal{K}}(\mu,\nu)} \mathbb{E}^\pi[c] - \varepsilon\,C + \varepsilon H(\pi^{\mathcal{K}}_{c,\varepsilon}(\mu,\nu)) &\le \inf_{\pi\in\Pi^{\mathcal{K}}(\mu,\nu)} \left\{\mathbb{E}^\pi[c] - \varepsilon\,H(\pi)\right\} + \varepsilon H(\pi^{\mathcal{K}}_{c,\varepsilon}(\mu,\nu)) \\
&\le \inf_{\pi\in\Pi^{\mathcal{K}}(\mu,\nu)} \mathbb{E}^\pi[c] + \varepsilon H(\pi^{\mathcal{K}}_{c,\varepsilon}(\mu,\nu)).
\end{aligned}
\tag{A.2}
$$

Now, note that $\inf_{\pi\in\Pi^{\mathcal{K}}(\mu,\nu)} \left\{\mathbb{E}^\pi[c] - \varepsilon\,H(\pi)\right\} = \mathcal{K}_{c,\varepsilon}(\mu,\nu) - \varepsilon H(\pi^{\mathcal{K}}_{c,\varepsilon}(\mu,\nu))$, and that, for $\varepsilon \to 0$, the LHS and RHS in (A.2) both tend to $\mathcal{K}_c(\mu,\nu)$. □

**Lemma A.2.** *Let $\mu$ and $\nu$ be discrete measures. Then*

$$\mathcal{K}_{c,\varepsilon}(\mu,\nu) \xrightarrow[\varepsilon\to\infty]{} \mathbb{E}^{\mu\otimes\nu}[c(x,y)].$$

*Proof.* Being $\mu$ and $\nu$ discrete, $\mathbb{E}^\pi[c]$ is uniformly bounded for $\pi \in \Pi^{\mathcal{K}}(\mu,\nu)$. Therefore, for $\varepsilon$ big enough, the optimizer in $\mathcal{P}^{\mathcal{K}}_{c,\varepsilon}(\mu,\nu)$ is $\hat{\pi} := \operatorname{argmax}_{\pi\in\Pi^{\mathcal{K}}(\mu,\nu)} H(\pi) = \mu \otimes \nu$, the independent coupling, for which $H(\mu \otimes \nu) = H(\mu) + H(\nu)$; see [14] and [22]. Therefore, for $\varepsilon$ big enough, we have $\mathcal{K}_{c,\varepsilon}(\mu,\nu) = \mathbb{E}^{\mu\otimes\nu}[c(x,y)]$. □

### A.2  Reformulation of the COT problem

*Proof.* The causal constraint (3.2) can be expressed using the following characteristic function:

$$
\sup_{l\in\mathcal{L}(\mu)} \mathbb{E}^\pi[l(x,y)] = \begin{cases} 0 & \text{if } \pi \text{ is causal;} \\ +\infty & \text{otherwise.} \end{cases}
\tag{A.3}
$$

This allows to rewrite (3.3) as

$$
\begin{aligned}
\mathcal{P}^{\mathcal{K}}_{c,\varepsilon}(\mu,\nu) &= \inf_{\pi\in\Pi(\mu,\nu)} \left\{ \mathbb{E}^\pi\left[c(x,y)\right] - \varepsilon H(\pi) + \sup_{l\in\mathcal{L}(\mu)} \mathbb{E}^\pi[l(x,y)] \right\} \\
&= \inf_{\pi\in\Pi(\mu,\nu)} \sup_{l\in\mathcal{L}(\mu)} \left\{ \mathbb{E}^\pi\left[c(x,y) + l(x,y)\right] - \varepsilon H(\pi) \right\} \\
&= \sup_{l\in\mathcal{L}(\mu)} \inf_{\pi\in\Pi(\mu,\nu)} \left\{ \mathbb{E}^\pi\left[c(x,y) + l(x,y)\right] - \varepsilon H(\pi) \right\} \\
&= \sup_{l\in\mathcal{L}(\mu)} \mathcal{P}_{c+l,\varepsilon}(\mu,\nu),
\end{aligned}
$$

where the third equality holds by the min-max theorem, thanks to convexity of $\mathcal{L}(\mu)$, and convexity and compactness of $\Pi(\mu,\nu)$. □

## A.3 Sinkhorn divergence at the level of mini-batches

**Empirical observation of the bias in Example 3.4.** In the experiment mentioned in Example 3.4, we consider a set of distributions $\nu$'s as sinusoids with random phase, frequency and amplitude. We let $\mu$ be one element in this set whose amplitude is uniformly distributed between minimum $0.3$ and maximum $0.8$. On the other hand, for each $\nu$, the amplitude is uniformly distributed between the same minimum $0.3$ and a maximum that lies in $\{0.4, 0.5, \ldots, 1.2\}$. Thus, the only parameter of the distribution being varied is the maximum amplitude. We may equivalently take the maximum amplitude as a single $\theta$ that parameterizes $\nu_\theta$, so that $\mu = \nu_{0.8}$. Figure 1 illustrates that the sample Sinkhorn divergence (3.7) (or regularized distance (2.4)) does not recover the optimizer $0.8$, while the proposed mixed Sinkhorn divergence (3.8) does.

**Comparison of various implementations.** Motivated by Bellemare et al. [10], Salimans et al. [37] address the problem of bias in the mini-batch gradients of Wasserstein distance by proposing a mini-batch Sinkhorn divergence that is closely related to (3.8). We denote the implementation of a mini-batch Sinkhorn divergence in Salimans et al. [37] as

$$\widehat{\mathcal{W}}^6_{c,\epsilon} := \mathcal{W}_{c,\varepsilon}(\hat{\mathbf{x}}, \hat{\mathbf{y}}_\theta) + \mathcal{W}_{c,\varepsilon}(\hat{\mathbf{x}}, \hat{\mathbf{y}}'_\theta) + \mathcal{W}_{c,\varepsilon}(\hat{\mathbf{x}}', \hat{\mathbf{y}}_\theta) + \mathcal{W}_{c,\varepsilon}(\hat{\mathbf{x}}', \hat{\mathbf{y}}'_\theta)$$
$$- 2\mathcal{W}_{c,\varepsilon}(\hat{\mathbf{x}}, \hat{\mathbf{x}}') - 2\mathcal{W}_{c,\varepsilon}(\hat{\mathbf{y}}_\theta, \hat{\mathbf{y}}'_\theta).$$

In addition to (3.7) and (3.8), we further consider other possible variations of the Sinkhorn divergence at the level of mini-batches, including

$$\widehat{\mathcal{W}}^3_{c,\epsilon} := 2\mathcal{W}_{c,\varepsilon}(\hat{\mathbf{x}}, \hat{\mathbf{y}}_\theta) - \mathcal{W}_{c,\varepsilon}(\hat{\mathbf{x}}, \hat{\mathbf{x}}') - \mathcal{W}_{c,\varepsilon}(\hat{\mathbf{y}}_\theta, \hat{\mathbf{y}}'_\theta)$$

and

$$\widehat{\mathcal{W}}^8_{c,\epsilon} := \mathcal{W}_{c,\varepsilon}(\hat{\mathbf{x}}, \hat{\mathbf{y}}_\theta) + \mathcal{W}_{c,\varepsilon}(\hat{\mathbf{x}}, \hat{\mathbf{y}}'_\theta) + \mathcal{W}_{c,\varepsilon}(\hat{\mathbf{x}}', \hat{\mathbf{y}}_\theta) + \mathcal{W}_{c,\varepsilon}(\hat{\mathbf{x}}', \hat{\mathbf{y}}'_\theta)$$
$$- \mathcal{W}_{c,\varepsilon}(\hat{\mathbf{x}}, \hat{\mathbf{x}}') - \mathcal{W}_{c,\varepsilon}(\hat{\mathbf{y}}_\theta, \hat{\mathbf{y}}'_\theta) - \mathcal{W}_{c,\varepsilon}(\hat{\mathbf{x}}, \hat{\mathbf{x}}) - \mathcal{W}_{c,\varepsilon}(\hat{\mathbf{y}}_\theta, \hat{\mathbf{y}}_\theta).$$

The superscripts in $\widehat{\mathcal{W}}^3_{c,\epsilon}$, $\widehat{\mathcal{W}}^6_{c,\epsilon}$ and $\widehat{\mathcal{W}}^8_{c,\epsilon}$ indicate the number of terms used in the mini-batch implementation of the Sinkhorn divergence. In the same spirit, our choice of mixed Sinkhorn divergence $\widehat{\mathcal{W}}^{\mathrm{mix}}_{c,\epsilon}$ corresponds to $\widehat{\mathcal{W}}^4_{c,\epsilon}$.

We compare the performance of all the variations in the low-dimensional applications of multivariate AR-1 and 1-D noisy oscillation (see Appendix B for experiment details) in Figure 6 and Figure 7, and in the high-dimensional applications of Sprite animations and the Weizmann Action dataset in Table 2. The superscripts on COT-GAN correspond to the Sinkhorn divergence used in the experiments. We replace the COT-GAN objective (3.8) with (3.7) in the experiment of COT-GAN$^2$, with $\widehat{\mathcal{W}}^3_{c,\epsilon}$ in COT-GAN$^3$, with $\widehat{\mathcal{W}}^6_{c,\epsilon}$ in COT-GAN$^6$, and with $\widehat{\mathcal{W}}^8_{c,\epsilon}$ in COT-GAN$^8$, respectively.

Figure 6: Results on learning the multivariate AR-1 process.

In the low-dimensional experiments, COT-GAN outperforms COT-GAN$^6$ on the 1-D noisy oscillation, but underperforms it on the multivariate AR-1 experiment. Both COT-GAN and COT-GAN$^6$ obtain

Figure 7: 1-D noisy oscillation. Top two rows show two samples from the data distribution and generators trained by different methods. Third row shows marginal distribution of pixels values (y-axis clipped at 0.07 for clarity). Bottom row shows joint distribution of the position of the oscillation at adjacent time steps.

significantly better results than all other variations of the mini-batch Sinkhorn divergence. Given the low-dimensional results, we only compare COT-GAN and COT-GAN[6] in the high-dimensional experiments. As shown in Table 2, COT-GAN performs the best in all evaluation metrics except for KVD for Sprites animation. Both COT-GAN and COT-GAN[6] perform better than MoCoGAN in these two tasks. However, because COT-GAN[6] requires more mini-batches in the computation, it is about 1.5 times slower than COT-GAN.

Table 2: Evaluations for video datasets. Lower value indicates better sample quality.

| **Sprites** | FVD | FID | KVD | KID |
|---|---|---|---|---|
| MoCoGAN | 1 108.2 | 280.25 | 146.8 | 0.34 |
| COT-GAN[6] | 620.1 | 109.1 | **64.5** | 0.091 |
| COT-GAN | **458.0** | **84.6** | 66.1 | **0.081** |
| **Human actions** | | | | |
| MoCoGAN | 1 034.3 | 151.3 | 89.0 | 0.26 |
| COT-GAN[6] | 630.8 | 109.2 | 46.79 | 0.19 |
| COT-GAN | **462.8** | **58.9** | **43.7** | **0.13** |

**The MMD limiting case.** In the limit $\varepsilon \to \infty$, Genevay et al. [21] showed that $\mathcal{W}_{c,\varepsilon}(\mu, \nu) \to \mathrm{MMD}_{-c}(\mu, \nu)$ under the kernel defined by $-c(x, y)$. Here we want to point out an interesting fact about the limiting behavior of the mixed Sinkhorn divergence.

**Remark A.3.** *Given distributions of mini-batches $\hat{\mathbf{x}}$ and $\hat{\mathbf{y}}$ formed by samples from $\mu$ and $\nu$, respectively, in the limit $\varepsilon \to \infty$, the Sinkhorn divergence $\widehat{\mathcal{W}}_{c,\varepsilon}(\hat{\mathbf{x}}, \hat{\mathbf{y}})$ converges to a biased estimator of $\mathrm{MMD}_{-c}(\mu, \nu)$; given additional $\hat{\mathbf{x}}'$ and $\hat{\mathbf{y}}'$ from $\mu$ and $\nu$, respectively, the mixed Sinkhorn divergence $\widehat{\mathcal{W}}_{c,\varepsilon}^{mix}(\hat{\mathbf{x}}, \hat{\mathbf{x}}', \hat{\mathbf{y}}, \hat{\mathbf{y}}')$ converges to an unbiased estimator of $\mathrm{MMD}_{-c}(\mu, \nu)$.*

*Proof.* The first part of the statement relies on the fact that $\mathrm{MMD}_{-c}(\hat{\mathbf{x}}, \hat{\mathbf{y}})$ is a biased estimator of $\mathrm{MMD}_{-c}(\mu, \nu)$. Indeed, we have

$$\widehat{\mathcal{W}}_{c,\varepsilon}(\hat{\mathbf{x}}, \hat{\mathbf{y}}) \xrightarrow{\varepsilon \to \infty} \mathrm{MMD}_{-c}(\hat{\mathbf{x}}, \hat{\mathbf{y}}) = -\frac{1}{m^2} \sum_{i=1}^{m} \sum_{j=1}^{m} [c(x^i, x^j) + c(y^i, y^j) - 2c(x^i, y^j)].$$

Now note that

$$\frac{1}{m^2}\sum_{i=1}^{m}\sum_{j=1}^{m}\mathbb{E}[c(x^i,x^j)] = \frac{1}{m^2}\left[\sum_{i=1}^{m}\mathbb{E}_{\mu}[c(x^i,x^i)] + \sum_{i\neq j}\mathbb{E}_{\mu\otimes\mu}[c(x^i,x^j)]\right]$$

$$= \frac{m-1}{m}\mathbb{E}_{\mu\otimes\mu}[c(x,x')],$$

where we have used the fact that $c(x^i,x^i)=0$. A similar result holds for the sum over $c(y^i,y^j)$. On the other hand, $\frac{1}{m^2}\sum_{ij}\mathbb{E}[c(x^i,y^j)] = \mathbb{E}_{\mu\otimes\nu}[c(x,y)]$. Therefore

$$\mathbb{E}\,\mathrm{MMD}_{-c}(\hat{\mathbf{x}},\hat{\mathbf{y}}) = -\frac{m-1}{m}[\mathbb{E}_{\mu\otimes\mu}[c(x,x')] + \mathbb{E}_{\nu\otimes\nu}[c(y,y')]] + 2\mathbb{E}_{\mu\otimes\nu}[c(x,y)]$$

$$\neq \mathrm{MMD}_{-c}(\mu,\nu),$$

which completes the proof of the first part of the statement.

For the second part, note that $\mathcal{W}_{c,\varepsilon}(\mu,\nu) \to \mathbb{E}_{\mu\otimes\mu}[c(x,x')]$ as $\varepsilon \to \infty$ [21, Theorem 1], thus

$$\widehat{\mathcal{W}}_{c,\varepsilon}^{\mathrm{mix}}(\hat{\mathbf{x}},\hat{\mathbf{x}}',\hat{\mathbf{y}},\hat{\mathbf{y}}') \to \mathbb{E}_{\hat{\mathbf{x}}\otimes\hat{\mathbf{y}}}[c(x,y)] + \mathbb{E}_{\hat{\mathbf{x}}'\otimes\hat{\mathbf{y}}'}[c(x',y')] - \mathbb{E}_{\hat{\mathbf{x}}\otimes\hat{\mathbf{x}}'}[c(x,x')] - \mathbb{E}_{\hat{\mathbf{y}}\otimes\hat{\mathbf{y}}'}[c(y,y')]$$

$$= \frac{1}{m^2}\sum_{i=1}^{m}\sum_{j=1}^{m}[c(x^i,y^i) + c(x'^i,y'^i) - c(x^i,x'^i) - c(y^i,y'^i)].$$

The RHS is an unbiased estimator of MMD, since its expectation is

$$\mathbb{E}_{\mu\otimes\nu}[c(x,y)] + \mathbb{E}_{\mu\otimes\nu}[c(x',y')] - \mathbb{E}_{\mu\otimes\mu}[c(x,x')] - \mathbb{E}_{\nu\otimes\nu}[c(y,y')] = \mathrm{MMD}_{-c}(\mu,\nu).$$

$\square$

The mixed divergence may still be a biased estimate of the true Sinkhorn divergence. However, in the experiment of Example 3.4 we note that the minimum is reached for the parameter $\theta$ close to the real one (Figure 1, bottom).

## B  Experimental details

### B.1  Low dimensional time series

Here we describe details of the experiments in Section 5.1.

**Autoregressive process.**  The generative process to obtain data $\mathbf{x}_t$ for the autoregressive process is

$$\mathbf{x}_t = \mathbf{A}\mathbf{x}_{t-1} + \boldsymbol{\zeta}_t, \quad \boldsymbol{\zeta}_t \overset{\mathrm{i.i.d}}{\sim} \mathcal{N}(0,\boldsymbol{\Sigma}), \quad \boldsymbol{\Sigma} = 0.5\mathbf{I} + 0.5,$$

where $\mathbf{A}$ is diagonal with ten values evenly spaced between 0.1 and 0.9. We initialize $\mathbf{x}_0$ from a 10-dimensional standard normal, and ignore the data in the first 10 time steps so that the data sequence begins with a more or less stationary distribution. We use $\lambda = 0.1$ and $\varepsilon = 10.0$ for this experiment.

**Noisy oscillation.**  This dataset comprises paths simulated from a noisy, nonlinear dynamical system. Each path is represented as a sequence of $d$-dimensional arrays, $T$ time steps long, and can be displayed as a $d \times T$-pixel image for visualization. At each discrete time step $t \in \{1,\ldots,T\}$, data at time $t$, given by $\mathbf{x}_t \in [0,1]^d$, is determined by the position of a "particle" following noisy, nonlinear dynamics. When shown as an image, each sample path appears visually as a "bump" travelling rightward, moving up and down in a zig-zag pattern as shown in Figure 8 (top left).

More precisely, the state of the particle at time $t$ is described by its position and velocity $\mathbf{s}_t = (s_{t,1},s_{t,2}) \in \mathbb{R}^2$, and evolves according to

$$\mathbf{s}_t = \mathbf{f}(\mathbf{s}_{t-1}) + \boldsymbol{\zeta}_t, \quad \boldsymbol{\zeta}_t = \mathcal{N}(0,0.1\mathbf{I}),$$

$$\mathbf{f}(\mathbf{s}_{t-1}) = c_t\mathbf{A}\mathbf{s}_{t-1}; \quad c_t = \frac{1}{\|\mathbf{s}_{t-1}\|_2 \exp(-4(\|\mathbf{s}_{t-1}\|_2 - 0.3) + 1)},$$

Figure 8: 1-D noisy oscillation. Same distributions as in 7 are shown.

where $\mathbf{A} \in \mathbb{R}^{2 \times 2}$ is a rotation matrix, and $\mathbf{s}_0$ is uniformly distributed on the unit circle.

We take $T = 48$ and $d = 20$ so that $\mathbf{x}_t$ is a vector of evaluations of a Gaussian function at 20 evenly spaced locations, and the peak of the Gaussian function follows the position of the particle $s_{t,1}$ for each $t$:

$$x_{t,i} = \exp\left[-\frac{(\mathrm{loc}(i) - s_{t,1})^2}{2 \times 0.3^2}\right],$$

where $\mathrm{loc} : \{1, \ldots, d\} \to \mathbb{R}$ maps pixel indices to a grid of evenly spaced points in the space of particle position. Thus, $\mathbf{x}_t$, the observation at time $t$, contains information about $s_{t,1}$ but not $s_{t,2}$. A similar data generating process was used in [43], inspired by Johnson et al. [26].

We compare the marginal distribution of the pixel values $x_{t,i}$ and joint distribution of the bump location ($\mathrm{argmax}_i x_{t,i}$) between adjacent time steps. See Figure 8.

**Electroencephalography.** We obtain EEG dataset from [45] and take the recordings of all the 43 subjects in the control group under the matching condition (S2). For each subject, we choose 75% of the trials as training data and the remaining for evaluation, giving 2 841 training sequences and 969 test sequences in total. All data are subtracted by channel-wise mean, divided by three times the channel-wise standard deviation, and then passed through a $\tanh$ nonlinearity. For COT-GAN, we train three variants corresponding to $\lambda$ being one of $\{1.0, 0.1, 0.01\}$, and $\varepsilon = 100.0$ for all OT-based methods. Data and samples are shown in Figure 9.

**Model and training parameters.** The dimensionality of the latent state is 10 at each time step, and there is also a 10-dimensional time-invariant latent state. The generator common to COT-GAN, direct minimization and SinkhornGAN comprise a 1-layer (synthetic) or 2-layer (EEG) LSTM networks, whose output at each time step is passed through two layers of fully connected ReLU networks. We used Adam for updating $\theta$ and $\varphi$, with learning rate 0.001. Batch size is 32 for all methods except for direct minimization of the mixed and original Sinkhorn divergence which is trained with batch size 64. These hyperparameters do not substantially affect the results.

The same discriminator architecture is used for both $h$ and $M$ in COT-GAN and the discriminator of the SinkhornGAN. This network has two layers of 1-D causal CNN with stride 1, filter length 5. Each layer has 32 (synthetic data) or 64 neurons (EEG) at each time step. The activation is ReLU except at the output which is linear for autoregressive process, sigmoid for noisy oscillation, and $\tanh$ for EEG.

For COT-GAN, $\lambda = 10.0$ and $\epsilon = 10$ for synthetic datasets, and $\lambda \in \{0.01, 0.1, 1.0\}$ and $\epsilon = 100.0$ for EEG. The choice of $\epsilon$ is made based on how fast it converges to a particular threshold of the transport plan, and each iteration takes around 1 second on a 2.6GHz Xeon CPU.

Figure 9: Data and samples obtained by different methods for EEG data, the number after COT-GAN indicates the value of $\lambda$.

## B.2 Videos datasets

### B.2.1 Sprite animations

**Data pre-processing.** The sprite sheets can be created and downloaded from [2]. The data can be generated with various feature options for clothing, hairstyle and skin color, etc. Combining all feature options gives us 6352 characters in total. Each character performs spellcast, walk, slash, shoot and hurt movements from different directions, making up to a total number of 21 actions. As the number of frames $T$ ranges from 6 to 13, we pad all actions to have the same length $T = 13$ by repeating previous movements in shorter sequences. We then crop the characters from sheets to be in the center of each frame, which gives a dimension of $64 \times 64 \times 4$ for each frame. We decide to drop the 4th color channel (alpha channel) to be consistent with the input setting of baseline models. Finally, the resulting dataset has 6352 data points consisting of sequences with 13 frames of dimensions $64 \times 64 \times 3$.

Table 3: Generator architecture.

| Generator | Configuration |
|---|---|
| Input | $z \sim \mathcal{N}(\mathbf{0}, \mathbf{I})$ |
| 0 | LSTM(state size = 128), BN |
| 1 | LSTM(state size = 256), BN |
| 2 | Dense(8*8*512), BN, LeakyReLU |
| 3 | reshape to 4D array of shape (m, 8, 8, 512) as input for DCONV |
| 4 | DCONV(N512, K5, S1, P=SAME), BN, LeakyReLU |
| 5 | DCONV(N256, K5, S2, P=SAME), BN, LeakyReLU |
| 6 | DCONV(N128, K5, S2, P=SAME), BN, LeakyReLU |
| 7 | DCONV(N3, K5, S2, P=SAME) |

Table 4: Discriminator architecture.

| Discriminator | Configuration |
|---|---|
| Input | 64x64x3 |
| 0 | CONV(N128, K5, S2, P=SAME), BN, LeakyReLU |
| 1 | CONV(N256, K5, S2, P=SAME), BN, LeakyReLU |
| 2 | CONV(N512, K5, S2, P=SAME), BN, LeakyReLU |
| 3 | reshape to 3D array of shape (m, T, -1) as input for LSTM |
| 4 | LSTM(state size = 512), BN |
| 5 | LSTM(state size = 128) |

#### B.2.2 The Weizmann Action database

**Data pre-processing.** The videos in this dataset consists of clips that have lengths from 2 to 7 seconds. Each second of the original videos contains 25 frames, each of which has dimension 144x180x3. To avoid the absence of objects at the beginning of the videos and to ensure an entire evolution of motions in each sequence, we skip the first 5 frames, then skip every 2 frames and collect 16 frames in a whole sequence as a result. Due to limited access to hardware, we also downscale each frame to $64 \times 64 \times 3$. The training set used contains 89 data points with dimensions $16 \times 64 \times 64 \times 3$.

**GAN architectures.** We detail the GAN architectures used in the experiment of the Weizmann Action database in Table 3 and Table 4. A latent variable $z$ of shape $5 \times 5$ per time step is sampled from a multivariate standard normal distribution and is then passed to a 2-layer LSTM to generate time-dependent features, followed by 4-layer deconvolutional neural network (DCONV) to map the features to frames. In order to connect two different types of networks, we map the features using a feedforward (dense) layer and reshape them to the desired shape for DCNN. In Table 3 and 4, the DCONV layers have N filter size, K kernel size, S strides and P padding option. We adopted batch-normalisation layers and the LeakyReLU activation function. We have two networks to parameterize the process $h$ and $M$ as discriminator share the same structure, shown in Table 4.

We use a fixed length $T = 16$ of LSTM. The state size in the last LSTM layer corresponds to the dimensions of $h_t$ and $M_t$, i.e., $j$ in (3.9). We also applied exponential decay to learning rate by $\eta_t = \eta_0 r^{s/c}$ where $\eta_0$ is the initial learning rate, $r$ is decay rate, $s$ is the current number of training steps and $c$ is the decaying frequency. In our experiments, we set the initial learning rate to be 0.001, decay rate 0.98, and decaying frequency 500. The batch size $m$ and time steps $T$ used are both 16. We have $\lambda = 0.01$, $\epsilon = 6.0$ and the Sinkhorn $L = 100$ in this experiment. We train COT-GAN on a single NVIDIA Tesla P100 GPU for 3 or 4 days. Each iteration takes roughly 1.5 seconds.

## C Sprites and human action results without cherry-picking

In this section we show random samples of Sprites and human actions generated by COT-GAN, mixed Sinkhorn minimization, and MoCoGAN without cherry-picking. The background was static for both experiments. In the Sprites experiments (see Figure 10), the samples from mixed Sinkhorn minimization and COT-GAN are both of good quality, whereas those from MoCoGAN only capture a rough pattern in the frames and fail to show a smooth evolution of motions.

Figure 10: Random samples with no cherry picking from models trained on animated Sprites. Top row: real sequences on the left and mixed Sinkhorn minimization on the right; bottom row: MoCoGAN on the left and COT-GAN on the right.

In Figure 11, we show a comparison of real and generated samples for human action sequences. Noticeable artifacts of COT-GAN and mixed Sinkhorn minimization results include blurriness and even disappearance of the person in a sequence, which normally happens when the clothing of the person has a similar color as the background. MoCoGAN also suffers from this issue and, visually, there appears to be some degree of mode collapse. We used generators of similar capacity across all models and trained COT-GAN, mixed Sinkhorn minimization and MoCoGAN for 65000 iterations.

Figure 11: Random samples with no cherry picking from models trained on human actions. Top row: real sequences on the left and mixed Sinkhorn minimization on the right; bottom row: MoCoGAN on the left and COT-GAN on the right.

## Footnotes

[2]Original dataset is available at `gaurav.munjal.us/Universal-LPC-Spritesheet-Character-Generator/` and `github.com/jrconway3/Universal-LPC-spritesheet`. To facilitate the use of large dataset in TensorFlow, we pre-shuffled all data used and wrote into tfrecord files. Links for download can be found on the Github repository.