[Reviews · NeurIPS 2020]

Review 1

Summary and Contributions: This paper proposes a framework for train GANs to produce data that is sequential in nature. The central ideas in this work are inspired by classical optimal transport theory; in particular, a probabilistic approach is taken in which the objective is to minimize the difference between the data distribution and the induced distribution by exploiting the notion of causal transport plans. Further, the authors propose a regularization scheme based on the entropy of the learned coupling. Through a variety of experiments, the authors show that their method generates realistic sequential data sequences and outperforms other competing methods

Strengths: + The formulation here is quite clean. The paper begins with the development and formulation of the causal optimal transport problem in (3.1), which has found applications in a variety of fields. This background is needed and helps set up the final formulation the authors propose for COT-GAN. + The experiments are impressive, both in their quality and in their breadth. Indeed it does seem that this method outperforms its peers and is capable of generating high quality data. The comparison between COT-GAN and MOCO-GAN in Figure 9 is quite impressive.

Weaknesses: - The paper is quite dense on pages 2-4. To facilitate broader appeal, it may be efficacious to given intuitive explanations for some of the theory that is developed. For instance, while the background up to problem 3.1 is written fairly clearly, it was difficult to build intuition for the set \mathcal{H}(\mu). An intuitive explanation for what this set is capturing (i.e. the set of tuples (h,M) where h is a "nice" -- continuous, bounded -- function and M is a martingale WRT to the filtration F^{\mathcal{X}}) would be useful. Sacrificing some of the theory (and moving it to the appendix) and adding more intuition may make things a bit clearer. - In Figure 4, the images are quite hard to see. Using fewer of the smaller cells may help. - Some comments on the stability of training these models seem to be missing. Given the well-documented instability of training these models, one wonders how difficult it is to get these models working in practice. Hopefully a full release of the code will alleviate some of these concerns. [EDIT: the authors will add a section on this topic in a future version, so I think my concern here has been addressed]

Correctness: For the most part all seems correct to me. However, I did have some confusion with one small point. In this work, the induced distribution is denoted by \mu, and the authors define it by \mu = \zeta \circ g^{-1}. This was a bit confusing to me. I may be mistaken here, but I would have thought that the induced distribution should

Clarity: While at times the paper is a bit dense, which is likely do to the spatial constraints, the paper is quite well written. The sections flow cleanly and the writing is of high-quality.

Relation to Prior Work: The prior work is adequately discussed.

Reproducibility: Yes

Additional Feedback: From my perspective, this was a very solid submission. The experiments are quite strong, the theory is solid, and the writing and presentation are very good. I enjoyed reading you work.


Review 2

Summary and Contributions: The paper introduces a novel class of Implicit Generative Model (IGM) that respects causality to better generate sequential data. Using the existing causal optimal transport theory, the paper formulates the Causal Optimal Transport GAN (COT-GAN) for this purpose. On a wide range of datasets, ranging from traditional time series to high dimensional videos, the approach outperforms baselines.

Strengths: The work is fairly principled in that it builds directly on the existing theory of Wasserstein GANs (WGANs) and causal optimal transport. The COT-GAN objective is novel and the requisite martingale condition is imposed in a fairly reasonable way. Section 3.3, bias reduction with the mixed Sinkhorn divergence, appears empirically useful. The empirical results are also generally strong. Figure 2 shows good performance when learning a conventional autoregressive process, and the evaluation for video datasets given in Table 1 shows that the method exceeds the MoCoGAN baseline by a very significant margin with respect to conventional quality metrics (FVD, FID, KVD, KID) on two video datasets.

Weaknesses: I have some concerns about novelty, in the sense that at a high level the paper is taking a pre-existing causal optimal transport framework and plugging it into a WGAN-like setting. However, as Section 3.4 mentions, this comes with some concerns that have to be resolved, although their resolution is not particularly arduous.

Correctness: The claims look reasonable (I skimmed the Appendix proofs for correctness, though did not thoroughly check) and the empirical methodology is correct; tables and figures also look sensible.

Clarity: The paper is fairly well-written; explanations are clear and easy to understand.

Relation to Prior Work: Relevant related work is discussed both in the early background sections (e.g. Section 2) and in the later Section 4 related work is explicitly discussed and novel contributions are delineated.

Reproducibility: Yes

Additional Feedback: Although the paper idea borrows heavily from pre-existing work (causal optimal transport, WGAN), their unification in COT-GAN requires some non-obvious concern resolution and produces a model that appears to work rather well in practice, as measured by three time series datasets and two video datasets. Hence, I lean towards acceptance. With respect to the method that reduced bias with mixed Sinkhorn divergence in Section 3.3, I would like to see a more theoretical justification. It seems like the motivation is primarily from intuition and the justification is currently empirical. [Post-rebuttal Update] I have reviewed the rebuttal; regarding novelty, my original review already highlighted that the bridge required the additional nontrivial work in e.g. Section 3.4, the question was only whether the additional work was significant enough. Upon further review, I have decided that it is, and I concede that the paper's strong results helped sway me. The paper will be a positive contribution to the conference.


Review 3

Summary and Contributions: This paper presents a a causal framework for Optimal-Transport based GANs. The idea is to rely on Genevay et al. [20] with two important modifications: (i) restrict the optimal transport problem to causal transport plans and (ii) correct the minibatch Sinkhorn estimation by relying on samples from two distinct batches at a time.

Strengths: This paper deals with an important problem, that is the generation of realistic sequential data. Both contributions listed above are sound and likely to have impact on the community. Also, experimental validation tends to confirm the theoretical strengths of these contributions.

Weaknesses: The discussion around the strategy used to correct mini-batch based estimation of Sinkhorn is very interesting, but somehow frustrating. Indeed, part of the discussion is pushed to supplementary material (cf. comment above) and even this supplementary material section ends with "We defer detailed analysis of mixed divergence to a future paper." This leaves us with a rather convincing toy experiment in which the corrected minibatch version seems to outperform its competitors in terms of its ability to retrieve optimal distribution parameters, but deeper understanding of the approach is somehow lacking.

Correctness: I found no specific issue in the theoretical claims. In the experiments, MoCoGAN is the only baseline for video generation: have the authors considered the use of a non video-specific baseline like TimeGAN (which is used as a baseline for other experiments)? Another concern regarding the experimental setup is that it would be interesting to have information about the performance of COT-GAN without the W^mix trick. [EDIT post-rebuttal] This last comment has been covered by additional experiments provided in the rebuttal.

Clarity: The paper is well written. Still, it is a real pity that the discussion in Appendix A.3 is not included in the main paper. Though authors are of course not responsible for strict page limits, this discussion is (in my opinion) among the most important parts of the paper (especially the inequality derived from triangular inequality and remark A.3) and would deserve some room in the core of the paper.

Relation to Prior Work: Yes.

Reproducibility: Yes

Additional Feedback:


Review 4

Summary and Contributions: The paper proposes COT-GAN, a specialized GAN for implicit generative modeling sequential data. The success of COT-GAN comes from Causal Optimal Transport (COT) and a new, low-bias divergence for better learning of such models. The authors showed the effectiveness of such methods under synthetic cases and some simple real-world cases.

Strengths: 1. Sufficient discussion of previous challenges in modeling sequential data. 2. Theory-driven method development and good practice-theory consistency.

Weaknesses: 1. The experiment is weak. To be honest the freedom in terms of independent dimensionality of the chosen data may not be very high, since these data have rather limited mode. This may leave the success of training the model on some rich-information dataset (like observation of particle interactions in physics research) questionable. (wrong suggestion) 2. To me the construction of the theory could use more illustration and intermediate experiments to further verify their conclusions. (maybe inappropriate suggestion) 3. Compared to TimeGAN, some important baselines, like WaveGAN is missing. (addressed)

Correctness: I tend to admit the correctness of the methods proposed in this paper.

Clarity: In general, yes.

Relation to Prior Work: To me the connection of the improved OT loss and the idea of using causal optimal transport are rather independent to each other. Still not sure to what extent this work could differ from previous similar attempts.

Reproducibility: Yes

Additional Feedback: After having had a discussion of the paper, I feel the rebuttal and other reviewers have convinced me about the good nature of the paper. However I still don't feel I have completely understood the paper and thus I would give a score of 7 with lower confidence 2.

[Author Response · NeurIPS 2020]

We thank all reviewers for the insightful feedback. Below we address all questions raised in the reviews. We will add the related discussions and further experiment results in the new version, shall our paper be accepted.

**[Reviewer 1]** • **Intuition.** More intuition can be added in Section 3. For example, the set $\mathcal{H}(\mu)$ mentioned by the reviewer is a class of test functions for causal transports. Intuitively: causality is a concept of conditional independence ($y_t$ independent of $x_{>t}$, given $x_{\leq t}$), that can be expressed in terms of conditional expectations, which in turn naturally leads to a formulation that involves martingales. • **Stability.** We can add more detail regarding stability of training COT-GAN. Empirically, we do observe that COT-GAN is relatively stable, in the sense that the loss tends to converge and that small changes in the hyperparameters do not obviously worsen the results especially in the lower dimensional settings. For high dimensional datasets, we indeed observe that sample quality fluctuates during training, which is a shortcoming suffered by all GANs. • **Figure 4.** We can show fewer frames.

**[Reviewer 2]** • **Novelty.** While most approaches rely on improving model architecture, compositional losses or domain-specific techniques, COT-GAN is a principled way of targeting generic sequential generation. Importantly, *causal* optimal transport was not present in the Machine Learning literature, and the proposed method is definitely not a mere marriage of two existing theories (COT and WGAN). First, it was not obvious to see that COT, formulated as a min-max optimization problem, naturally falls into the GAN framework. And even after this bridge was created, the development of the algorithm required substantial effort. Given our positive results, we believe the new theory of COT could greatly benefit sequential learning. • **Justification for mixed-Sinkhorn.** Our choice of mixed-Sinkhorn is inspired by the idea of taking into account the variation within the distributions $\mu$ and $\nu$, as a way to correct the bias originating by mini-batched training. To support our intuition, we provide two arguments in Appendix A.3: the triangular inequality and the convergence to an unbiased estimator. Furthermore, this is confirmed empirically via the experiments in the paper as well as the additional results in the Figure below.

**[Reviewer 3]** • **Discussion on mixed-Sinkhorn.** We are happy to move some discussion to the main body of the paper. For the justification, please see our response to Reviewer 2. • **Comparison to other baselines.** We can add more details in the paper. In the Figure below, we provide an extra comparison between COT-GAN and TimeGAN, WaveGAN (trained with WGAN-GP loss) and COT-GAN without the mixing trick. Combined with the results in the paper, we see that the mixing trick is critical for the success of training and that COT-GAN achieves the best results among all.

**[Reviewer 4]** • **Weak Experiments.** We respectfully disagree with the reviewer on this comment. We thoroughly demonstrated the results for low and high dimensional applications using a variety of well-established performance metrics. Related work mostly focuses on either low or high dimensional datasets but not both, and often lacks reports on basic statistical features of the generated samples. For example, we achieved good results on EEG data without any domain-specific modifications, which outperform similar work specifically targeting EEG. As for the efficacy of our method on "rich-information datasets", we do not have reason to believe (either in theory or empirically) that our method would not be successful given sufficient computational resources. • **Comparison to other baselines.** Please see our response to Reviewer 3. • **Intermediate experiments.** It is unclear to us what is meant by "intermediate" experiments. If the reviewer is referring to experiments which investigate the specific contribution of each component of the model, we can include that in a later version of the paper. For example, isolating the impact of the martingale penalization $p_M$ can be achieved by modifying the value of $\lambda$ in (3.10), see Figure 8. We have run additional experiments omitting $p_M$ and the mixing trick for the AR dataset, see the Figure below. • **Connection of the improved OT loss with COT.** In our model, the class of loss functions (parametrized in (3.9)) over which D optimizes is the one emerging from (3.4), which is the representation of the (regularized) COT problem as optimization over (regularized) classical OT problems. Therefore, D is de facto calculating the *causal* distance between the original data and the generated one. For concerns about differentiation from previous attempts, please see our response regarding novelty to Reviewer 2.



[Meta-Review · NeurIPS 2020]

As the authors will notice, the rebuttal had a substantial positive impact on the reviews -- thanks to the authors for having crafted a compelling rebuttal. The authors should now revise the paper using reviews and add the material discussed in reviews / rebuttal (e.g. part on stability).